

# Study on the diagnostic potential and molecular mechanism of hsa_circ_0000831 in oral squamous cell carcinoma

Ting Liu[1,*], Xiaoze Chen[1,*], Shigeng Lin[2], Qitao Wen[2], Wei Deng[2] and Daiying Huang[3]

[1] School of Stomatology, Hainan Medical University, Haikou, China
[2] Department of Oral and Maxillofacial Surgery, Hainan Affiliated Hospital of Hainan Medical University, Hainan General Hospital, Haikou, China
[3] Department of Oral and Maxillofacial Surgery, The First Affiliated Hospital, Sun Yat-sen University, Guangzhou, China
[*] These authors contributed equally to this work.

## ABSTRACT

**Background**. Circular RNA (circRNA) are a new class of non-coding RNAs that are involved in the molecular pathology of cancer. This study aims to screen and validate key circRNAs with diagnostic potential in o ral squamous cell carcinoma (OSCC), and explore their possible molecular mechanism.

**Methods**. This study first integrated the GSE131182 dataset with clinically obtained OSCC sample data and used the limma package to identify differentially expressed circular RNAs (DEcircRNAs). Subsequently, circRNAs associated with head and neck squamous cell carcinoma were identified using CircNet2.0 and intersected with the differentially expressed circRNAs to determine the key circRNA. The diagnostic value of the key circRNA was evaluated using receiver operating characteristic (ROC) curves, followed by functional validation through *in vitro* assays including cell counting kit-8 (CCK-8), wound healing, transwell assay, and flow cytometry. Finally, target microRNAs (miRNAs) were predicted using CircNet2.0 and miRDB, a ceRNA network was constructed, and functional enrichment analysis of target genes was performed using Metascape tool.

**Results**. A total of 318 and 46 differentially expressed circRNAs (DEcircRNAs) were identified from the GSE131182 dataset and clinical samples, respectively. Through intersection analysis, the key circRNA hsa_circ_0000831 was identified. hsa_circ_0000831 was upregulated in OSCC samples, and ROC curve analysis indicated its high diagnostic performance. *In vitro* experiments showed that inhibition of hsa_circ_0000831 significantly reduced OSCC cell viability, migration, and invasion, while markedly enhancing apoptosis. ceRNA network analysis predicted that hsa_circ_0000831 targets five miRNAs (including hsa-miR-136-5p, hsa-miR-100-3p, hsa-miR-144-5p, hsa-miR-149-5p and hsa-miR-214-5p), with the associated target genes mainly enriched in cancer-related pathways.

**Conclusion**. This work offer s a novel foundation for the early identification of OSCC and provides potential clues for finding new therapeutic targets.

Corresponding authors
Wei Deng, dengweisums2@126.com
Daiying Huang, daiying1966@126.com

# INTRODUCTION

Oral cancer is a prevalent malignancy, and oral squamous cell carcinoma (OSCC) is the most common pathological type, accounting for approximately 90% of all oral cancer cases worldwide (*Lan et al., 2024*; *Li, Tang & Dai, 2023*). The major risk factors for oral cancer include tobacco use, alcohol consumption, and betel nut chewing (*Li, Tang & Dai, 2023*; *Jiang et al., 2019*; *Dong, Zhang & Chang, 2023*). Ulceration is a typical clinical manifestation of OSCC (*Tan et al., 2023*), and nearly half of the patients are diagnosed at an advanced stage, making early detection challenging (*Li et al., 2025*). Currently, the primary treatments for OSCC involve surgery combined with radiotherapy and chemotherapy (*Gamez et al., 2018*). However, due to frequent tumor metastasis and recurrence (*Ling, Cheng & Tao, 2021*), the prognosis for OSCC remains poor, with a five-year survival rate of less than 50% (*Qi et al., 2025*; *Ge et al., 2015*). Therefore, the identification of reliable biomarkers is crucial for early diagnosis and may contribute to improving the survival rate of OSCC patients (*Seyfinejad & Jouyban, 2022*).

Circular RNA (circRNA) refers to a category of single-stranded non-coding RNAs that are covalently closed RNA molecules produced by reverse splicing (*Patop, Wüst & Kadener, 2019*; *Zeng et al., 2023*) and can modulate gene expression *via* acting as protein scaffolds and sponges and microRNAs (miRNAs) sponges, coding for proteins, and regulating splicing and transcription (*Fernández-Tussy, Ruz-Maldonado & Fernández-Hernando, 2021*; *Chen et al., 2023a*). The circRNAs are implicated in multiple types of carcinomas, exhibiting aberrant expression (*Zhang et al., 2018*; *Chen & Shan, 2021*; *Xu et al., 2023*). Previous research has manifested that circRNAs such as hsa_circ_0001971, hsa_circ_0001874, and circ-KIAA0907 promote the proliferation of OSCC cells by affecting multiple signaling pathways (*Dong et al., 2021*; *Jun et al., 2021*). CircFNDC3B has been found to accelerate angiogenesis and metastasis in OSCC (*Li et al., 2023*). Additionally, it has been discovered by researchers that circRNF13 facilitates the stabilization of ITGB1 messenger RNA (mRNA) through phase separation mediated by IGF2BP1 in a manner dependent on m6A, thereby contributing to the chemoresistance of oral cancer to cisplatin (*Xu et al., 2025*). Exploring the relationship between circRNAs and OSCC can help find new biomarkers and provide novel targets for early diagnosis of OSCC.

Therefore, in this study, we innovatively identified a circRNA, hsa_circ_0000831, as a potential biomarker for OSCC by integrating self-measured clinical data with publicly available datasets. Its functional role was subsequently validated through a series of *in vitro* experiments. Based on this biomarker, a ceRNA regulatory network involving associated miRNAs and target genes was constructed. The detection of this circRNA is expected to facilitate the earlier diagnosis of OSCC, thereby increasing the chances of timely intervention, improving clinical outcomes, and ultimately enhancing patient prognosis and quality of life.

## MATERIAL AND METHODS

### Obtaining of tissue samples from OSCC patients

OSCC patients who underwent surgeries in the Department of Oral and Maxillofacial Surgery of Hainan Provincial People's Hospital from November 2023 to April 2024 were collected. Three pairs of OSCC and adjacent normal tissues were acquired. After the surgically excised tissue specimens are removed from the body, they are washed with normal saline and then placed in a −80 °C refrigerator within half an hour. Inclusion criteria included patients without chemotherapy or radiotherapy before surgery, and tissue specimens that were pathologically confirmed as oral squamous cell carcinoma. Patients with cancer in other parts of the body were excluded. The medical ethics committee of Hainan General Hospital approved the current study (ethic approval no. [2024] 240). Written informed consent was acquired from the patients before the operation.

### RNA extraction

Following the specifications, total RNA was collected with the use of Trizol reagent (15596018CN, Thermofisher, Waltham, MA). Bioanalyzer 2100 and RNA 6000 Nano LabChip Kit (5067-1511, Agilent, CA, USA) was used to measure the quantity and purity of RNA, with RIN number > 7.0. About five μg of RNA was utilized to deplete ribosomal RNA through the Ribo-Zero Gold rRNA Removal Kit (MRZG12324, Illumina, San Diego, USA). Afterwards, employing divalent cations, the remanent RNAs were split into short fragments at a high temperature. Thereafter, reverse transcription of the RNA fragments into cDNA was achieved with the use of SuperScript™ II Reverse Transcriptase (1896649, Invitrogen, USA). Next, the U-labeled second-stranded DNAs was synthesized by E. coli DNA polymerase I (m0209, NEB, USA), RNase H (NEB, cat.m0297, USA) and dUTP Solution (R0133, Thermo Fisher, USA). A-base was added to the blunt ends of each strand to ligate to the indexed adapters that contained a T-base overhang. The fragments were then ligated with either single- or dual-index adapters, followed by size selection (300–600 bp) using the AMPureXP beads. The ligated products were amplified by polymerase chain reaction (PCR) after the U-labeled second-stranded DNAs were treated with the heat-labile UDG enzyme (m0280, NEB, USA). Specifically, the following conditions were set to carry out PCR amplification: initial denaturation at 95 °C for 3 min; 8 cycles at 98 °C for 15 s, at 60 °C for 15 s, at 72 °C for 30 s, and at 72 °C for 5 min. The cDNA library had an average insert size of 300 ± 50 bp. The final libraries were subjected to paired-end 150 bp sequencing (2 × 150 bp, PE150) on the Illumina NovaSeq 6000 platform (LC-Bio Technology Co., Ltd., Hangzhou, China). The sequencing depth per sample exceeded 20 million reads. Quality control of raw reads was performed using FastQC (v0.11.9), and reads with Phred quality score >30 for more than 85% of bases were retained for downstream analysis. The overall Q30 percentage and GC content were reported to ensure high-quality sequencing data.

### CircRNA identification of self-measured datasets

Circular RNAs were *de novo* assembled from mapped reads using CIRCExplorer2 (2.2.6, default) and CIRI (2.0.2, default). Next, the tophat-fusion and CIRCExplorer2 or CIRI

were employed to identify back-spliced junction reads from unmapped reads. This ensured the identification of unique circRNAs from the samples.

## Quantification of circRNA abundance

To quantitatively compare back splicing from different RNA-seq, the back-spliced reads (support for circRNA) were normalized through read length and number of mapped reads (spliced reads per billion mapping, SRPBM). The number of reads mapped to the reference genome in the samples was quantified, followed by normalizing the number of reads spanning the back-spliced junction to the total number of mapped reads (units in billion) and read length. The Perl script was utilized to calculate the SRPBM value for circRNA in each sample, which served as the abundance of expression. SRPBM = number of circular reads/number of mapped reads (units in billion)/read length.

## Data sources

The dataset used in this study includes public datasets and self-measured datasets obtained according to the previous methods. The public dataset was the GSE131182, which contained the circRNA expression profiles of six pairs of OSCC and normal oral mucosal tissue samples, was acquired from the Gene Expression Omnibus (GEO) database (https://www.ncbi.nlm.nih.gov/geo/query/acc.cgi?acc=GSE131182).

## Differential analysis

Employing the limma package (*Ritchie et al., 2015*), the differentially expressed circular RNAs (DEcircRNAs) in the GSE131182 and self-measured data were screened. DEcircRNAs were identified with the threshold of $|\log_2^{\text{FoldChange}}| \geq 1$ and $p < 0.05$. To control for false positives, multiple testing correction was performed using the Benjamini–Hochberg method, and adjusted $p$-values (FDR) were also calculated.

## Enrichment analysis

Parental genes for circRNAs up-regulated in OSCC were subjected to Gene Ontology (GO) enrichment analysis in three categories (biological process, cellular component, and molecular function) and Kyoto Encyclopedia of Genes and Genomes (KEGG) pathway analysis using the "clusterProfiler" R package (*Wang et al., 2025*).

## Identification of key circRNAs

The circRNAs with top50 associated with head and neck squamous cell carcinoma (HNSC) were identified using CircNet2.0 (https://awi.cuhk.edu.cn/~CircNet/php/index.php), after removing duplicates, and then intersected with the public dataset and the up-regulated circRNAs in the self-measured data to obtain the key circRNAs. The diagnostic sensitivity and specificity of the key circRNAs were verified using receiver operating characteristic (ROC) curves.

## Construction of competing endogenous RNA (ceRNA) network

The target miRNAs of key circRNAs were obtained by taking intersections through CircNet2.0 website and miRDB (https://mirdb.org/). The regulatory network was visualized by Ctyoscape 3.9.1 software (*Shannon et al., 2003*). Functional enrichment

analysis of target genes was conducted utilizing the Metascape online tool (https://metascape.org/gp/index.html#/main/step1).

## Cell culture

Human OSCC WSU-HN30 (IM-H729) and HSC-3 (IM-H705) were obtained from Immocell (http://www.immocell.com/) (Xiamen, China). All cells used were identified by short tandem repeats (STR) to confirm that they were free of contamination. Cultures were conducted using Dulbecco's Modified Eagle Medium (DMEM) (BDBio, Hangzhou, China) and Eagle's Minimum Essential Medium (EMEM) (30-2003, ATCC, Manassas, VA), respectively, with 10% fetal bovine serum (FBS) (C0226, Beyotime, Shanghai, China) and 1% penicillin-streptomycin (P/S) (15140148, ThermoFisher Scientific, Waltham, MA). The incubation was performed in an environment at 37 °C, 5% $CO_2$ and saturated humidity.

## Cell transfection and quantitative reverse transcription PCR (qRT-PCR)

The WSU-HN30 and HSC-3 cells were transfected with the hsa_circ_0000831 interference plasmid employing the Lipofectamine 2000 Transfection Kit (11668027, Invitrogen, Carlsbad, CA, USA). The plasmids (sh-hsa_circ_0000831 and sh-NC) were synthesized and constructed by GenePharma Co., Ltd. (Shanghai, China), and sequence verification was performed prior to transfection. The targeting sequence was 5′-GCTGTTTCTACACTTGCTAGG-3′, and negative control was set up. And sh-NC represents the negative control, while sh-hsa_circ_0000831 represents the WSU-HN30 or HSC-3 cells transfected with hsa_circ_0000831. After transfection, the cells were incubated at 37 °C for 48 h. After extracting RNA following a similar procedure as described previously, direct reverse transcription was performed. After generating cDNA, quantitative reverse transcription (qRT-PCR) was applied to test the transfection efficiency. The primer pairs were as follows: hsa_circ_0000831-F 5′-CGAGGTATAGCAGAAGAATCA-3′, hsa_circ_0000831-R 5′-CTTGGTTCAGCATCACTCT-3′; *GAPDH*-F 5′-CTACATGGTTTACATGTTCC-3′, *GAPDH*-R 5′-CATACTTCTCATGGTTCACA-3′. Afterwards, qRT-PCR was carried out applying AriaMx Real-Time PCR System (Agilent, USA) utilizing the SYBR Green I (SY1020, Solarbio). The expression level of hsa_circ_0000831 was obtained through $2^{-\Delta\Delta CT}$ method (*Zhang & Li, 2021*), and *GAPDH* was utilized for normalization.

## Cell counting Kit-8 (CCK-8) assay

Cell proliferation was examined using the Cell counting Kit-8 (CCK-8, C0037, Beyotime, Shanghai, China) according to the manufacturer's instructions. Transfected WSU-HN30 and HSC-3 cells were seeded into 96-well plates at a density of $5 \times 10^3$ cells per well in 100 µL of complete medium. Cells were incubated at 37 °C in a humidified incubator containing 5% $CO_2$. At 24, 48, and 72 h after seeding, 10 µL of CCK-8 reagent was added to each well, followed by incubation at 37 °C for 30 min while protected from light. After incubation, absorbance (OD) at 450 nm was detected in each well by an enzyme labeling instrument (Skanlt RE 7.0) (*Ma et al., 2023*).

### Wound healing assay

The transfected WSU-HN30 and HSC-3 cells were planted into 6-well plates respectively. After reaching full confluence, a sterile pipette tip was employed to make a straight scratch on the monolayer cells for creating an artificial wound. Next, after removing the used medium, Phosphate Buffered Saline (PBS) (C0221A, Beyotime, Shanghai, China) was used for washing the cells twice to discard cell debris. Then, the culture dishes were returned to the incubator after the addition of fresh serum-free medium under the original culture conditions for further incubation. The culture dishes were taken out at 0 h and 48 h respectively, and the scratched areas were photographed using a microscope (Olympus Corporation, Tokyo, Japan). The wound width changes were analyzed by ImageJ software, and the migration capability of WSU-HN30 and HSC-3 cells was reflected by wound closure rated at each time point measured.

### Transwell assay

The polycarbonate membrane (pore size: eight μm, 3422, Corning, Inc, Corning, NY, USA) of a 24-well plate was pre-coated with Matrigel (C0372, Beyotime, Shanghai, China). The upper transwell chamber had 200 μL of serum-free medium, whereas the lower transwell chamber contained 700 μL of medium and 10% FBS. After 48 h, 4% paraformaldehyde (P0099, Beyotime, Shanghai, China) was employed to fix the cells invading into the lower chamber, and 0.1% crystal violet (C0121, Beyotime, Shanghai, China) was used for dyeing the cells for 30 min. The cells were observed and quantified from three random fields of the inverted optical microscope (Olympus Corporation, Tokyo, Japan) to evaluate the invasive ability of WSU-HN30 and HSC-3 cells.

### Flow cytometry

After digesting the transfected WSU-HN30 and HSC-3 cells with 0.25% trypsin and cell washing with PBS (C0221A, Beyotime, China), Annexin V-FITC Apoptosis Detection Kit (C1062S, Beyotime) was employed to dye the cells, followed by using a flow cytometer (BD Biosciences, San Jose, CA, USA) for the detection of apoptosis.

### Statistical analysis

All the data were analyzed by R 4.2.0 software and GraphPad Prism 8. For two variables, the Unpaired t test was used for statistical test. Analysis of variance (ANOVA) and Sidak's multiple comparisons test were employed for three or more variables. Mean $\pm$ standard deviation (SD) was used to express the data. Differences in expression between OSCC and control samples were compared using the wilcoxon rank-sum test. And $p < 0.05$ was considered statistically significant. All cell experiments were performed in three independent replicates ($n = 3$).

## RESULTS

### Differential analysis of circRNAs

Differential analysis of circRNA expression profiles in 6 pairs of OSCC and normal oral mucosal tissue samples from GSE131182 detected a sum of 318 DEcircRNAs, including

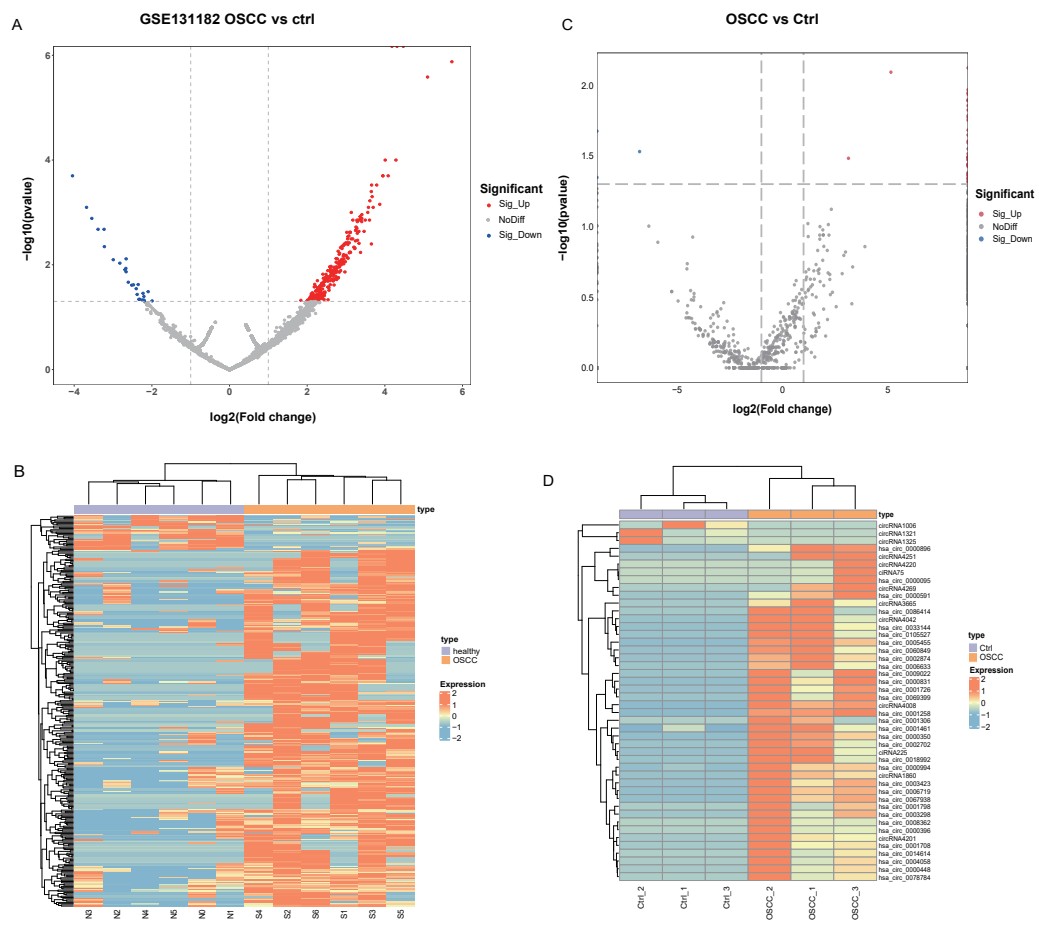

**Figure 1** **Identification of DEcircRNA.** (A) Volcano plot of DEcircRNA in GSE131182. (B) Heatmap of DEcircRNA in GSE131182. (C) Volcano plot of DEcircRNA in the self-measured data. (D) Heatmap of DEcircRNA in the self-measured data.

290 up-regulated and 28 down-regulated DEcircRNAs. The 318 statistically significant DEcircRNAs were displayed in the volcano plot (Fig. 1A) and clustered heatmap (Fig. 1B). Moreover, a sum of 46 DEcircRNAs were detected in the self-measured data, of which 43 were up-regulated and three were down-regulated, and displaying 46 DEcircRNAs (Fig. 1C, Fig. 1D).

## Enrichment analysis of parental genes for upregulated circRNAs

Since up-regulated circRNAs accounted for a majority of DEcircRNAs, enrichment analysis of parental genes for circRNAs upregulated in OSCC was performed. As revealed by Kyoto Encyclopedia of Genes and Genomes (KEGG) enrichment analysis, the up-regulated genes in GSE131182 were primarily implicated in pathways such as proteoglycans in cancer, regulation of actin cytoskeleton, endocytosis, *etc* (Fig. 2A). The Gene Ontology analysis in Molecular Function term displayed that the up-regulated genes in GSE131182 were chiefly relevant to the pathways of adenosine triphosphate (ATP) hydrolysis activity, protein serine/threonine kinase activity, and actin binding (Fig. 2B). The Cellular Component (CC)

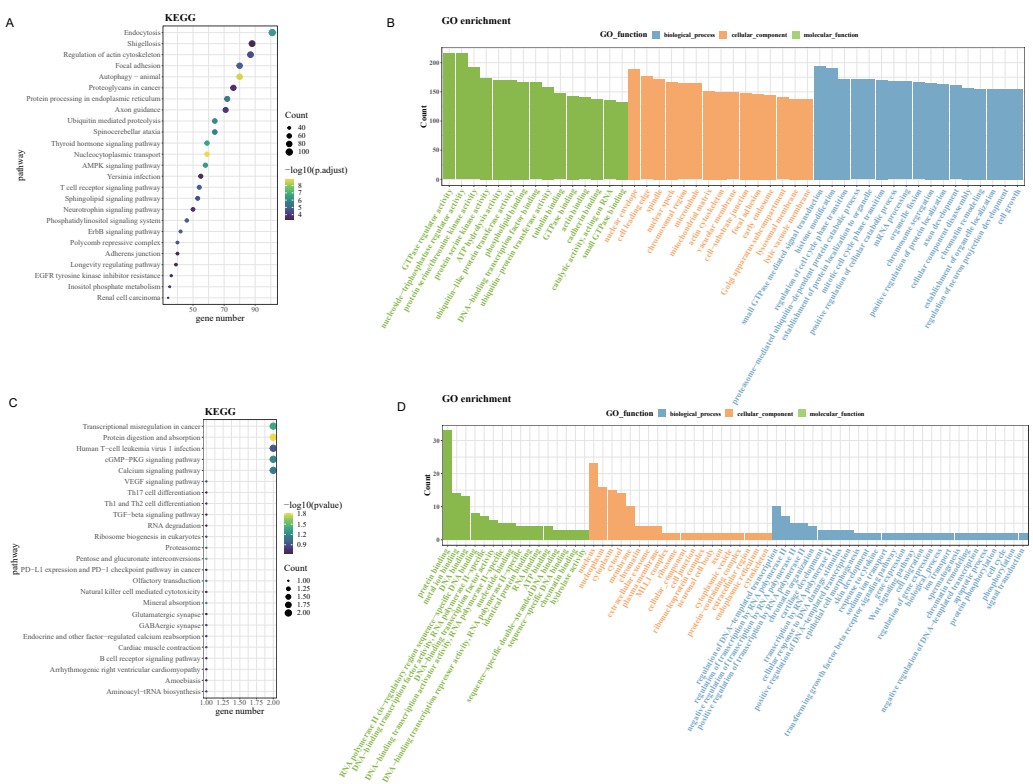

**Figure 2 Enrichment analysis.** (A) Results of KEGG enrichment analysis on the upregulated genes in GSE131182. (B) Upregulated genes in GSE131182 were subjected to GO-enrichment analysis. (C) Upregulated genes in self-measured data were subjected to KEGG enrichment analysis. (D) Upregulated genes in self-measured data were subjected to GO-enrichment analysis.

enrichment of GO showed that GSE131182 upregulated genes mainly in actin cytoskeleton and lytic vacuole membrane and mitochondrial matrix pathways (Fig. 2B). And Biological Process (BP) enrichment of GO demonstrated that the upregulated genes in GSE131182 were principally in the chromatin remodeling, proteasome-mediated ubiquitin-dependent protein catabolic process, and other pathways (Fig. 2B). While KEGG analysis of the self-measured data showed enrichment in calcium signaling pathway, human T-cell leukemia virus 1 infection, cGMP-PKG signaling pathway, transcriptional misregulation in cancer, protein digestion and absorption (Fig. 2C). GO analysis on the self-measured data revealed that the BP, CC and MF processes were respectively enriched in pathways such as protein binding, chromatin, and protein phosphorylation, among many others (Fig. 2D).

## Identification and validation of the biomarker

After removing duplicates from the top 50 circRNAs related to HNSC and then intersected with the up-regulated circRNAs from the public dataset and self-test data to obtain a circRNA: hsa_circ_0000831 (derived from the exon of CEP192) (Fig. 3A). Next, the expression of hsa_circ_0000831 in control samples and OSCC was compared in the two datasets. The hsa_circ_0000831 expression in the OSCC group of GSE131182 dataset had
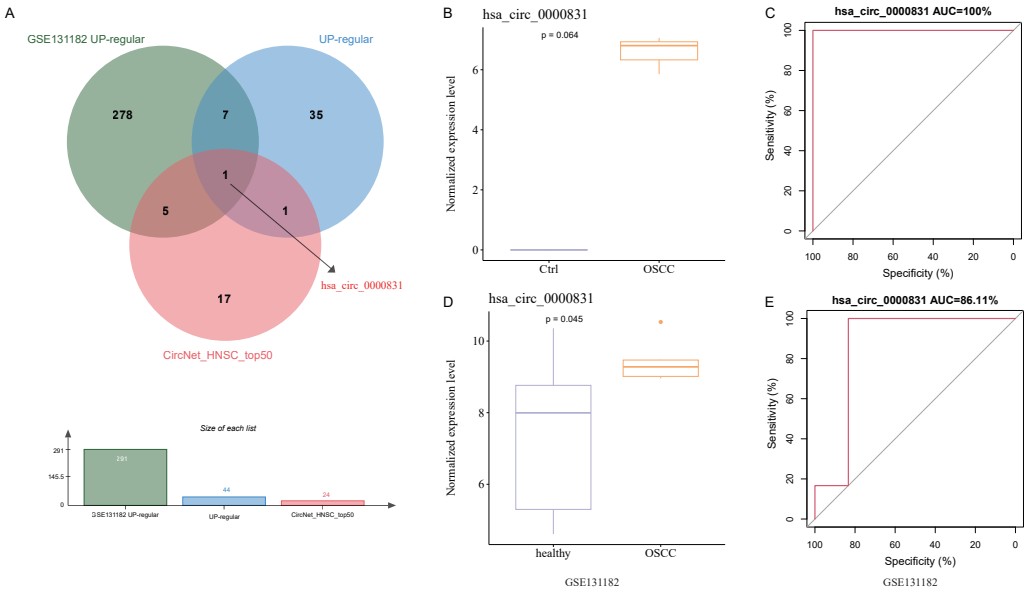

**Figure 3  Identification and validation of the key circRNA.** (A) Venn diagram for identification of the key circRNA. (B) Expression of hsa_circ_0000831 in GSE131182. (C) ROC curve of hsa_circ_0000831 in GSE131182 . (D) Expression of hsa_circ_0000831 in self-measured data. (E) ROC curve of hsa_circ_0000831 in self-measured data.

a trend to be higher than the control group (Fig. 3B), and its diagnostic value reflected in the area under the curve (AUC) value (Fig. 3C). The self-measured data also showed hsa_circ_0000831 was high-expressed in OSCC (Fig. 3D, $p < 0.05$), and the level of hsa_circ_0000831 expression was used to plot the ROC curve, which could be seen that the AUC was also large, implying higher diagnostic sensitivity and specificity (Fig. 3E). Thus, hsa_circ_0000831 might be able to be served as a diagnostic biomarker for OSCC.

## Results of *in vitro* experiments

Negative control and hsa_circ_0000831 were transfected into WSU-HN30 and HSC-3 cells and labeled as sh-NC and sh-hsa_circ_0000831, respectively. The results showed that the transfection was successful (Figs. 4A–4B). CCK8 assay displayed that the cell viability of the sh-NC group in WSU-HN30 and HSC-3 cells was notably higher than that of the sh-hsa_circ_0000831 group ($p < 0.05$, Figs. 4C–4D), indicating that the cell viability of OSCC decreased significantly after the inhibition of hsa_circ_0000831. The wound-healing assay indicated that the percentage of wound closure in the sh-NC group was remarkably higher than in the sh-hsa_circ_0000831 group ($p < 0.05$, Fig. 5A), implying that the migration ability of OSCC decreased significantly after the inhibition of hsa_circ_0000831. Transwell experiment displayed that the invasive capability of OSCC in the sh-NC group was observably higher than sh-hsa_circ_0000831 group ($p < 0.05$, Fig. 5B), indicating that the invasion ability of OSCC decreased significantly after the inhibition of hsa_circ_0000831. The results of flow cytometry suggested that the apoptosis of sh-NC group was signally lower than that of sh-hsa_circ_0000831 group ($p < 0.05$,

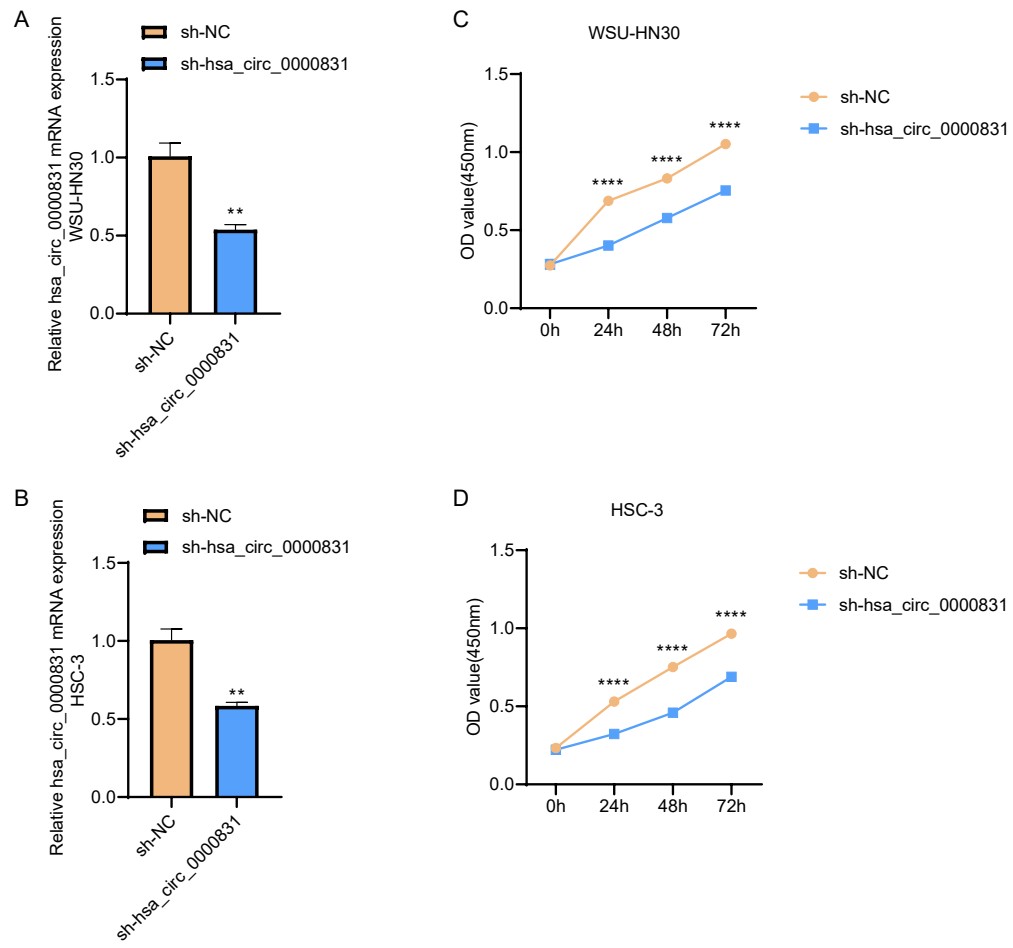

**Figure 4** **Cell transfection and CCK - 8 experiment.** (A) The transfection situation of WSU-HN30 cells. (B) The transfection situation of HSC-3 cells. (C) CCK-8 assay on WSU-HN30 cells. (D) HSC-3 cells were subjected to CCK-8 assay. **** means $p < 0.0001$; ** means $p < 0.01$.

Fig. 5C), indicating that the apoptosis ability of OSCC was remarkably elevated after the inhibition of hsa_circ_0000831.

## Analysis of ceRNA network and target gene enrichment

Given that circRNAs may serve as molecular sponges to lift the inhibition of miRNAs on mRNAs, therefore influencing mRNA expression. In this study, we took the top 5 miRNAs predicted to match hsa_circ_0000831 through CircNet 2.0 website, which were hsa-miR-136-5p, hsa-miR-100-3p, hsa-miR-144-5p, hsa-miR-149-5p and hsa-miR-214-5p, and the corresponding target genes were obtained by taking the intersection of CircNet and miRDB (Fig. 6A). Enrichment analysis suggested that these target genes were principally linked to regulation of secretion, regulation of T cell cytokine production, cellular responses to stimuli, and other cancer pathways (Fig. 6B).

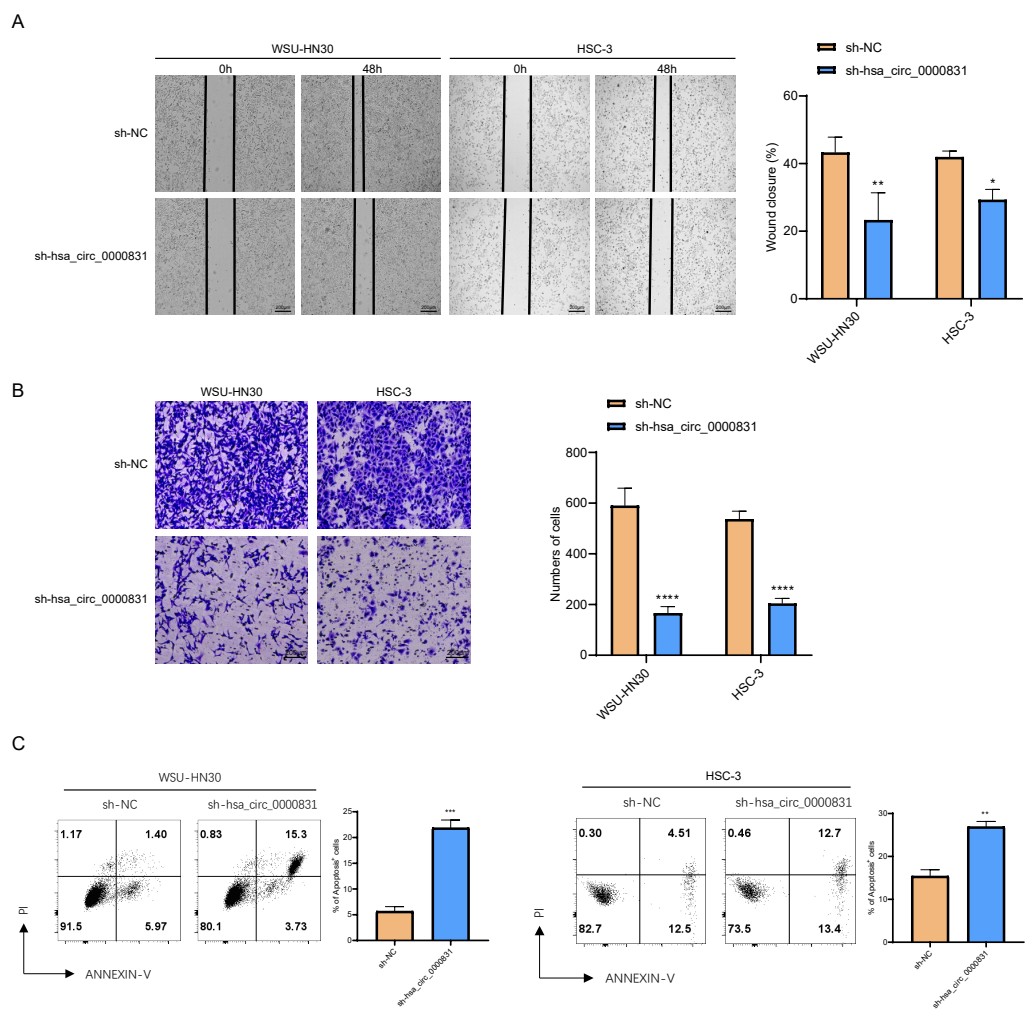

**Figure 5** **The *in vitro* experimental results.** (A) Wound healing assay s of WSU-HN30 and HSC-3. (B) Transwell assay s of WSU-HN30 and HSC-3. (C) Flow cytometry of WSU-HN30 and HSC-3. **** means $p < 0.0001$; *** means $p < 0.001$; ** means $p < 0.01$ ; * means $p < 0.0 5$.

## DISCUSSION

OSCC is a common malignant HNSC, which is mostly found in an advanced-stage and exhibits a terrible prognosis (*Ge et al., 2015*). The circRNAs are covalently closed-loop single-stranded RNAs that bind to miRNAs or other molecules to modulate gene expression at the level of transcription or post-transcription (*Chen et al., 2023b*). Research showed that circRNAs may exert an imperative effect in different types of cancers and serve as biomarkers for cancer (*Patop & Kadener, 2018*). Based on this, we discovered a circRNA named hsa_circ_0000831 as a biomarker for OSCC. Besides, we constructed a ceRNA network map and performed enrichment analysis on the target genes corresponding to the top 5 miRNAs matched by hsa_circ_0000831. These findings not only reveal the diagnostic

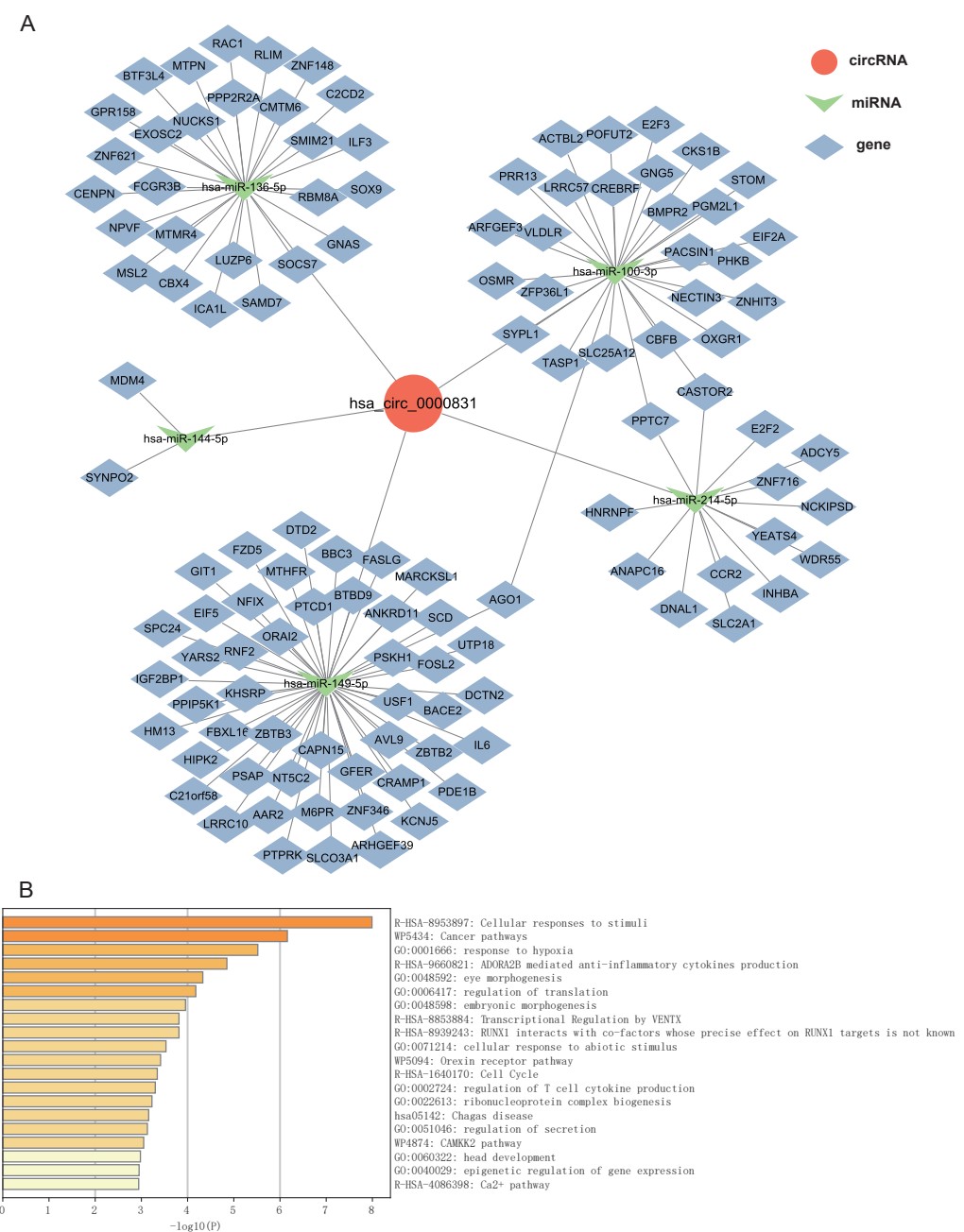

**Figure 6** **The ceRNA network prediction and enrichment analysis.** (A) The ceRNA network predicted downstream miRNAs and mRNAs of hsa_circ_0000831. (B) Functional enrichment analysis of downstream target genes of hsa_circ_0000831.

potential and regulatory mechanisms of hsa_circ_0000831 in OSCC, but also provide new molecular evidence for early screening and targeted therapy.

In this study, a circRNA, hsa_circ_0000831, derived from the *CEP192* gene (*Chen et al., 2022*), was identified and found to be significantly upregulated in OSCC tissues

based on both public datasets and clinical samples. Functional validation through *in vitro* assays demonstrated that knockdown of hsa_circ_0000831 markedly suppressed OSCC cell viability, migration, and invasion, while promoting apoptosis. Recently, *CEP192* has been recognized as a new gene involved in the advancement of non-alcoholic fatty liver disease (NAFLD) to hepatocellular carcinoma (HCC) (*Cai, Song & Yu, 2020*). In addition, *Liu et al. (2022)* found that *CEP192* expression was closely associated with an immunosuppressive tumor microenvironment and low immune phenotype scores, making it a potential predictor of HCC immune checkpoint inhibitor response. RNA sequencing studies had indicated that hsa_circ_0000831 expression was down-regulated in human HCC tissues (*Luo et al., 2022*). Taken together, these findings suggest that hsa_circ_0000831 may serve as a critical oncogenic regulator in OSCC by promoting malignant cell behaviors and potentially modulating immune-related pathways through its parental gene *CEP192*. The pathogenesis of many diseases, including OSCC, involves the dysregulated circRNA-miRNA-mRNA interaction network (*Sakshi et al., 2021*; *Saikishore et al., 2020*). By competing for shared miRNAs, ceRNAs regulate each other at the level of post-transcription (*Qi et al., 2015*). In this study, by building a network map of ceRNAs, we found that hsa-miR-214-5p, hsa-miR-136-5p, hsa-miR-149-5p, hsa-miR-100-3p, hsa-miR-144-5p act as miRNAs of hsa_circ_0000831, which may play a role in OSCC role. Previous studies have found that these miRNAs play roles in various diseases. For example, hsa-miR-136-5p, which can be sponged by circRNAs, has been implicated in androgenetic alopecia as well as several types of cancer (*Wei et al., 2022*; *Chee et al., 2025*), hsa-miR-100-3p is associated with proliferation, DNA synthesis, and apoptosis of human testicular support cells (*Liu et al., 2021*), hsa-miR-144-5p affects overall survival of patients with renal clear cell carcinoma (*Zhan et al., 2021*), hsa-miR-214- 5 p regulates tumor proliferation and migration in osteosarcoma (*Shi et al., 2020*), and endogenous hsa-miR-149-5p in spongiotic macrophages affects abdominal aortic aneurysms (AAA) by promoting IL-6 transcription and inflammatory cytokine secretion (*Ma et al., 2022*). These evidences laterally imply that hsa_circ_0000831 may affect OSCC development through these 5 miRNAs, but their specific relationship required additional experiments.

Functional enrichment analysis of hsa_circ_0000831 downstream target genes suggested that it was primarily enriched in several related pathways. Among them, the enrichment of cancer pathways further implied that hsa_circ_0000831 affects the occurrence of OSCC through circRNA-miRNA-mRNA interaction (*Li et al., 2021*). Whereas the enrichment of regulation of T cell cytokine production and regulation of secretion implied that this process was associated with immune response and cell secretion. the T cell immune response was closely related to the occurrence of OSCC (*Burassakarn et al., 2021*). Additionally, PD-1 secreted by T cells has been shown to be an important target for immune checkpoint blockade (ICB) therapy in OSCC (*Yu et al., 2024*).

The present study contained several limitations to be pointed out. First, the sample size used in this study was small, which may have led to individual differences or biases, limiting the generalizability of the results. Future studies will expand the clinical sample size and introduce OSCC tissue samples from multiple centers and regions for validation to improve the statistical power and representativeness of the study. In addition, the ceRNA

network construction is mainly based on bioinformatics predictions and has not yet been verified by molecular biology experiments. In the future, the direct binding relationship between hsa_circ_0000831 and predicted miRNAs can be verified by methods such as dual luciferase reporter assays and RNA pull-down, and the regulatory role of downstream target genes can be further confirmed. Finally, the current functional experiments have only been conducted in the OSCC cell line, and its role in animal models or clinical tissues has not yet been evaluated. Therefore, in subsequent studies, we will construct an OSCC mouse xenograft model to validate the role of hsa_circ_0000831 in tumorigenesis, metastasis, and treatment response *in vivo*.

## CONCLUSION

In this study, hsa_circ_0000831 was successfully identified based on clinical sample data and public databases. Its role in OSCC was confirmed through bioinformatics analysis and *in vitro* experiments. This finding provides a novel basis for the early detection of OSCC, which may improve diagnostic accuracy and allow timely intervention. Furthermore, it offers new insights into the molecular mechanisms of OSCC and potential therapeutic targets, paving the way for future research and the development of more effective treatment strategies.

**Abbreviations**

| | |
|---|---|
| **OSCC** | oral squamous cell carcinoma |
| **GCO** | Global Cancer Observatory |
| **circRNA** | circular RNA |
| **miRNA** | microRNA |
| **mRNA** | messenger RNA |
| **GEO** | Gene Expression Omnibus |
| **DEcircRNA** | differentially expressed circular RNA |
| **BP** | Biological Process |
| **CC** | Cellular Component |
| **MF** | Molecular Function |
| **GO** | Gene Ontology |
| **KEGG** | Kyoto Encyclopedia of Genes and Genomes |
| **ANOVA** | Analysis of Variance |

### Funding

The study was supported by Hainan Province Science and Technology Special Fund (ZDYF2024SHFZ098). The funders had no role in study design, data collection and analysis, decision to publish, or preparation of the manuscript.

### Grant Disclosures

The following grant information was disclosed by the authors:
Hainan Province Science and Technology Special Fund: ZDYF2024SHFZ098.

## Competing Interests

The authors declare there are no competing interests.

## Author Contributions

- Ting Liu performed the experiments, analyzed the data, prepared figures and/or tables, authored or reviewed drafts of the article, and approved the final draft.
- Xiaoze Chen performed the experiments, analyzed the data, authored or reviewed drafts of the article, and approved the final draft.
- Shigeng Lin performed the experiments, analyzed the data, prepared figures and/or tables, and approved the final draft.
- Qitao Wen performed the experiments, analyzed the data, prepared figures and/or tables, and approved the final draft.
- Wei Deng conceived and designed the experiments, authored or reviewed drafts of the article, and approved the final draft.
- Daiying Huang conceived and designed the experiments, prepared figures and/or tables, authored or reviewed drafts of the article, and approved the final draft.

## Human Ethics

The following information was supplied relating to ethical approvals (i.e., approving body and any reference numbers):

The medical ethics committee of Hainan General Hospital approved the current study.

## DNA Deposition

The following information was supplied regarding the deposition of DNA sequences:

The self-measured raw sequence data are available at the Genome Sequence Archive in National Genomics Data Center, China National Center for Bioinformation/Beijing Institute of Genomics, Chinese Academy of Sciences, GSA-Human: HRA011520.

## Data Availability

All raw data is available at Github and Zenodo:

- https://github.com/WeiDeng123/Raw-data.git
- WeiDeng123. (2025). WeiDeng123/Raw-data: Raw data (data). Zenodo. https://doi.org/10.5281/zenodo.15368971

## Supplemental Information

Supplemental information for this article can be found online at http://dx.doi.org/10.7717/peerj.20082#supplemental-information.

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
