# Peer review of "Study on the diagnostic potential and molecular mechanism of hsa_circ_0000831 in oral squamous cell carcinoma"

_PeerJ, doi:10.7717/peerj.20082_

## Round 0.1 · original submission · Major Revisions

Based on the reviewers' comments, I recommend a major revision of your manuscript. While the reviewers acknowledge the potential value of your work, they have raised several substantive concerns that need to be addressed. Please carefully consider and respond to all comments from both reviewers. In particular, focus on strengthening the methodological aspects and providing additional experimental evidence where suggested. A point-by-point response to all reviewers' comments should be included with your revised manuscript. Upon resubmission, your manuscript will be sent back to the reviewers for further evaluation.

**PeerJ Staff Note**: Please ensure that all review, editorial, and staff comments are addressed in a response letter and that any edits or clarifications mentioned in the letter are also inserted into the revised manuscript where appropriate.

**PeerJ Staff Note**: It is PeerJ policy that additional references suggested during the peer-review process should only be included if the authors agree that they are relevant and useful.

**Language Note**: The review process has identified that the English language must be improved. PeerJ can provide language editing services - please contact us at [email protected] for pricing (be sure to provide your manuscript number and title). Alternatively, you should make your own arrangements to improve the language quality and provide details in your response letter. – PeerJ Staff

Reviewer 1 ·

Basic reporting

There are no improvements to add.

Experimental design

There are no improvements to add.

Validity of the findings

There are no improvements to add.

Additional comments

In future studies, Digital PCR (dPCR) can be used to determine the quantity of DNA or RNA with high sensitivity and absolute quantification.

Reviewer 2 ·

Basic reporting

This study identified hsa_circ_0000831 as a key circRNA in oral squamous cell carcinoma (OSCC) through bioinformatics analysis and experimental validation. Inhibiting this circRNA suppressed cell proliferation, migration, and invasion while promoting apoptosis. miRNA target prediction and functional analysis revealed potential regulatory mechanisms, providing new insights for early diagnosis and therapeutic strategies in OSCC. However, there are still some deficiencies in details in the manuscript. Please revise it carefully according to the comments below.
1. Line 21-23, The purpose of this article should be clarified. Line 24-34, The description in the methods section is rather disorganized. Line 53-40, The results section is not fully described. It is recommended to review and complete it.
2. Line 47-74, Although the epidemiology, pathogenic factors, and treatment status of OSCC are comprehensively reviewed, the text should be condensed to avoid redundancy (e.g., in sections on incidence and risk factors) and improve logical flow and readability. The study’s innovation should also be clearly stated in the introduction, such as the first systematic identification and validation of a specific circRNA as a potential OSCC biomarker and the construction of its ceRNA regulatory network.
3. In the Material and methods section, the specific R packages and methods with parameters have been supplemented. For example, what R package was used for the enrichment analysis.
4. Line 98-108, It is recommended to include specific parameters for RNA library construction (e.g., insert size range, library concentration measurement method) and sequencing quality metrics (e.g., Q30 ≥ 90%) to enhance reproducibility.
5. Line 133, When using the limma package for differential analysis, the p-value threshold is set to < 0.05, but it is not mentioned whether FDR (False Discovery Rate) correction (such as the Benjamini-Hochberg method) was performed. It is recommended to include relevant explanations to avoid false positive results.
6. The discussion section does not sufficiently integrate the experimental results, such as the observed decreases in cell proliferation, migration, and invasion, along with increased apoptosis following hsa_circ_0000831 knockdown. hsa_circ_0000831 is derived from the CEP192 gene, but the function of this gene itself in the cell cycle or cancer was not mentioned in the discussion. It should further elaborate on the potential molecular mechanisms underlying these findings.
7. The GO/KEGG enrichment results revealed multiple immune-related pathways (such as T-cell cytokine generation and secretion regulation), but failed to link these pathways with the tumor microenvironment or immunotherapy of OSCC for in-depth discussion.
8. The roles of multiple mirnas in other diseases are cited in the article, but it is not clear whether these mirnas have been reported or functionally verified in OSCC.
9. There are some grammatical errors, inappropriate word usage or redundant expressions such as "manifested to be implicated" and "spongy hsa-miR-136-5p", as well as complex sentence structures that affect readability. Sentence of "could offer new biomarkers for the early identification of OSCC" are rather arbitrary and lack sufficient restrictions on the limitations of the research.
10. Although the small sample size and the lack of mechanism validation were mentioned, it was not explained how these limitations affected the generalisation of the conclusion or the direction for future improvement. For example, has the use of independent cohorts been considered to verify the diagnostic efficacy? Is there a plan to establish animal models to evaluate in vivo functions?

Experimental design

Please see the Basic reporting.

Validity of the findings

Please see the Basic reporting.

Reviewer 3 ·

Basic reporting

This study reveals the cancer-promoting role of hsa_circ_0000831 in OSCC for the first time, and provides a new candidate marker for early diagnosis. Despite the limitations of sample size and mechanism verification, its idea of integrating multi-omics analysis and experimental verification is innovative. It is suggested that the authors supplement and improve the methodological details, and further explore the regulatory mechanism of ceRNA network, so as to improve the clinical transformation value of research. The following are my specific opinions.
1、The overall language of the manuscript is clear, but sentences need simplification. The writing should be more concise and polished.
2、Pay attention to the abbreviations in the keywords. For example, check whether "hsa_circ_0000831" follows standard naming conventions and is easily searchable. Also, evaluate if "in vitro experiment" is an appropriate keyword.
3、The background section should include recent research advances (from the past three years) on circRNAs in oral squamous cell carcinoma (OSCC).
4、The number of replicates for in vitro experiments should be increased to at least n≥3.
5、The descriptions of CCK-8 and flow cytometry experimental procedures are too brief. Please supplement them with more details or references to established protocols.
6、In section 2.11, provide information about the source of the plasmids, such as the company name and other relevant details.
7、The discussion only generally mentions "limited sample size" without specifying how this may lead to biases.
8、The font size of labels in some volcano plots and heatmaps is too small, affecting the visual quality. This needs to be adjusted.
9、The literature citations are outdated. Try to cite more recent studies (from the past three years) to improve the timeliness of the references.

Experimental design

no comment

Validity of the findings

no comment

---

## Round 0.2 · accepted · Accept

Both reviewers are satisfied with your revisions and have recommended acceptance. I am happy to let you know that your manuscript will be published.

Reviewer 2 ·

Basic reporting

Accept revisions. No further comments.

Experimental design

Accept revisions. No further comments.

Validity of the findings

Accept revisions. No further comments.

Reviewer 3 ·

Basic reporting

The manuscript is ready for publication.

Experimental design

The manuscript is ready for publication.

Validity of the findings

The manuscript is ready for publication.